# Sociodemographic intersections and risk of multiple long-term conditions: A systematic review

**Mansuk Daniel Han** [1]*, **Thomas Yates**[2,3], **Kamlesh Khunti**[1,2], **Cameron Razieh**[1,2,4], **Francesco Zaccardi**[1,2]

**1** Leicester Real World Evidence Unit, Diabetes Research Centre, University of Leicester, United Kingdom, **2** Diabetes Research Centre, University of Leicester, Leicester, United Kingdom, **3** National Institute for Health and Care Research, Leicester Biomedical Research Centre, University of Leicester, Leicester, United Kingdom, **4** Office for National Statistics, Newport, United Kingdom

* mdh26@leicester.ac.uk

## Abstract

Multimorbidity, or multiple long-term conditions (MLTC), is a growing public health concern with implications for quality of life, healthcare utilisation, and premature mortality. Classical explanations for MLTC often treat sociodemographic categories as independent predictors, overlooking the relational dynamics of health inequalities. This systematic review examines how MLTC outcomes vary at the intersections of sociodemographic factors within their relational context. We conducted a systematic search of PubMed, Medline, and Scopus to identify 792 studies. Four studies met inclusion criteria but none were longitudinal, which limits our ability to examine the role of intersectional effects on MLTC outcomes over the life course from this review. A narrative synthesis was conducted due to their wide heterogeneity among the MLTC outcomes of the studies included in this review. The limited evidence may potentially suggest that MLTC outcomes can vary considerably at the intersections of sociodemographic factors. All four studies in this review suggested that the association of income with MLTC outcomes can vary by what other sociodemographic factors it intersects with. The role of disability on MLTC outcomes varied when intersected with ethnicity, at least in the US racial context. A low level of education is a known MLTC risk factor, but when intersected with ethnicity for both men and women in the South African setting, definitive cumulative disadvantages were not found in the projected life expectancy. Future intersectionality-informed quantitative MLTC research should prioritise using longitudinal data and solution-linked variables to inform context-responsive interventions.

**Data availability statement:** All relevant data are within the paper and its Supporting Information files.

**Funding:** The author(s) received no specific funding for this work.

**Competing interests:** The authors have declared that no competing interests exist in relation to the content of this paper.

## Introduction

Multimorbidity, or multiple long-term conditions (MLTC), is defined as the simultaneous presence of two or more long-term conditions (LTC) in an individual [1]. MLTC reduces quality of life, ability to work, and life expectancy, while increasing hospitalisation risk and disproportionate use of health and social care resources [2], compounding pressure on individuals and social systems alike. Due to the lack of standard measurement definition, the burden of MLTC can vary greatly. A meta-analysis from 2022 found the pooled prevalence of MLTC at 42.4% (95% CI 38.9% to 46.0%), with 47.8% of the high heterogeneity attributable to study participant mean age and the number of conditions included for defining MLTC [3].

MLTC is often viewed as an inevitable byproduct of increasing lifespan [4] but it is actually a complex biological, psychological, and social phenomenon [5] from multifaceted risk exposures [6,7]. For example, a cross-sectional study of 1.16 million people from the UK Clinical Practice Research Datalink (CPRD) revealed wide disparities in MLTC prevalence when accounting for area-level deprivation [1]. In the most deprived group, the prevalence of mental-physical MLTC was 34.3% among those aged 45–49 years. It was matched in the least deprived group by among those aged 85–89 years at 31.1%, confirming previous findings from Scotland [8].

Based on the social determinants of health (SDH) theory [9–12], various socio-epidemiological hypotheses on MLTC development [13] conjecture that non-modifiable and modifiable risk factors [5,14] are interlinked. Factors such as sociodemographic background [15,16] and genetic predisposition [14] cannot be altered by interventions or behavioural changes. Lifestyle and behavioural attributes [14,17] are considered modifiable risk factors and intervention targets, such as stress, sleep, physical activity, smoking, or diet [18]. However, interventions targeting modifiable factors often work best with people of higher socioeconomic status [19,20], because the structural constraints of socio-environmental disadvantages shape behavioural choices [21–23]. Conversely, SDH-informed policy interventions targeting non-modifiable factors often struggle to demonstrate local-level impact [24] because they inherently treat sociodemographic factors as mutually isolated predictors [25] or deficit indicators [26]. It leads to conflating or neglecting important intragroup variation [27] for marginalised populations.

Intersectionality theory offers an alternative approach to inform context-responsive interventions that are relevant to specific communities [28]. Rooted in Black feminist legal theory by Kimberlé Crenshaw [29], intersectionality theory shifts the focus from the SDH view of social categories as separate and static entities to understanding them as interconnected [30] and relational [31] systems of power. While originally qualitative, recent methodological advances enable operationalising intersectionality in population health research [28,32–35], making it a viable analytic tool for MLTC equity research. However, being in nascent stages [35,36], the application of intersectionality has been mostly limited to studying single diseases. It is unclear whether and how intersecting sociodemographic positions and identities shape MLTC outcomes.

The aim of this systematic review is to examine how the interplay of sociodemographic factors plays a role on MLTC outcomes by their relational context. We identify

and summarise quantitative studies that examine MLTC outcomes through an intersectionality lens; assess which social positions (e.g., gender, ethnicity, income) have been studied in combination; and discuss how their interplay in MLTC outcomes varies when considered within an intersectional context.

## Methods

### Eligibility criteria

Inclusion criteria included: quantitative studies on MLTC (longitudinal or cross-sectional) published in peer-reviewed journals (in any language), which define MLTC as a combination of two or more recognised diseases/conditions; define intersecting sociodemographic factors as exposure; and MLTC prevalence, incidence, or health outcomes as the outcome of interest.

Exclusion criteria included: single-disease studies; qualitative studies; expert opinion or committee reports; studies which define MLTC as a combination of symptoms or pre-disease conditions, that is, not defined in ICD-10 diseases (e.g., pre-disease, frailty, disability, quality of life); measure transitions or trajectories within a single disease (e.g., cancer metastasis and/or advancing in stages); and those published in grey literature.

The inclusion and exclusion criteria are summarised in Table 1.

### Information sources and search strategy

A comprehensive search was conducted across PubMed, Medline, and Scopus on 16 April 2025, to identify studies from database inception. The search strategy combined terms for MLTC and intersections of sociodemographic factors: *(Intersect\*) AND (multimorbidity OR multiple long term condition\* OR MLTC OR multiple chronic condition\* OR syndemic).* Note that the term "intersectionality" here refers to the specific sociological definition, which implies *"interconnections of social categorisations such as race, class, and gender, creating overlapping and interdependent systems of discrimination or disadvantage"* [37].

This systematic review has been registered with PROSPERO (CRD420251006288) and conducted and reported in line with the PRISMA statement [38] (S1 Table for the PRISMA 2020 checklist).

### Study selection

The first author and a second reviewer (FZ) screened each record retrieved by the following selection process. After removing duplicates, based on the pre-defined inclusion and exclusion criteria, the identified articles were screened in the following order: (a) whether a study is a population-based study; (b) whether a study is a quantitative study; (c) whether a study defines MLTC as an outcome variable; and (d) whether a study defines intersectionality of sociodemographic factors (e.g., ethnicity, sex/gender, deprivation, disability, etc.) as exposure variables. The remaining studies were examined by the author and the second reviewer for extracting data and determining synthesis eligibility.

**Table 1. Inclusion and exclusion criteria.**

| | Inclusion | Exclusion |
|---|---|---|
| **Study design** | Quantitative studies on MLTC (longitudinal or cross-sectional) | Single-disease studies, Qualitative studies, Expert opinion/committee reports |
| **Methodology** | Intersecting sociodemographic factors as exposure, MLTC status as outcome(s), Measure MLTC trajectories longitudinally within the same individuals or cross-sectionally, MLTC defined as a combination of recognised diseases/conditions (e.g., self-report, ICD-9, ICD-10) | MLTC defined as a combination of symptoms or pre-disease conditions, that is, not defined in ICD-10 diseases (e.g., pre-disease, frailty, disability, quality of life), Transitions or trajectories within a single disease (e.g., cancer metastasis or advancing in stages) |
| **Publication** | Peer-reviewed journal articles (in any language) | Grey literature |

## Study eligibility for synthesis

The studies which passed the initial screening were further scrutinised for synthesis eligibility if they reported on MLTC development, progression (including mortality) in those with prevalent MLTCs, or severity outcomes, and included inter-sectional analyses of sociodemographic factors.

Particular attention was paid to whether the studies applied intersectionality analysis approaches correctly. Because the use of intersectionality approach in quantitative health research is in nascent stages [33,35,39], the claims of applying intersectionality warrant case-by-case examination. Therefore, we examined each study by the criteria for improving quantitative intersectional research suggested by Bauer and colleagues [32].

## Study characteristics

The first author (MDH) and a second reviewer (FZ) screened each record for study population and design, outcome variables, independent variables (exposure), reference group, intersectional analysis method, data source, definition for MLTC, list of LTCs, and setting. Particular attention was paid to participant sociodemographic variables and their inter-secting permutations pertaining to intersectional analysis. The analytical method of each study was assessed against the categories of intersectionality approaches in health research by Guan and colleagues [35].

## Outcome variables

The primary MLTC outcome domains sought were such as prevalence, mortality, life expectancy, hospitalisation, or number of LTC accumulated.. For each study, all results compatible with MLTC outcome domains were sought, including results from various statistical models, time points, and subgroups. If studies presented multiple measures for the same outcome domain, the most comprehensive or methodologically robust analyses (e.g., adjusted models over crude comparisons) were prioritised.

## Outcome measures

For each study, relevant measures reflecting both single-categorical and intersectional nature (i.e., odds, risk, rate, hazard ratios; absolute risks) were reviewed.

## Study Risk of Bias Assessment

The Newcastle-Ottawa Scale (NOS) tool was used to assess the risk of bias for each study reviewed. The assessment was conducted manually, supported by a NOS template [40] for recording judgments.

## Synthesis of Results

It was not possible to pooling the results in a meta-analysis due to the wide heterogeneity of outcome measures between studies. The outcome measures of the two Spanish studies [41,42] were MLTC prevalence and all-cause 5-year mortality. The outcome measure of the US study [43] were MLTC-related emergency visits, MLTC-related hospitalisation, and MLTC-related deaths. The outcome measures of the South African study [31] were modelled projections of overall life expectancy and multimorbid life expectancy. Therefore, a narrative synthesis was carried out to summarise the findings instead.

# Results

## Study selection

From a total of 792 records identified from PubMed (n=474), Medline (n=293), and Scopus (n=25), duplicates (n=309) were removed. Then, based on the pre-defined inclusion and exclusion criteria, the remaining 483 records were screened: (1) whether a study is mainly an MLTC study (402 records excluded); (2) whether a study is a quantitative study (53

records excluded); (3) whether a study defines MLTC as an outcome variable (13 records excluded); and (4) whether a study applies intersecting sociodemographic factors as exposure (10 records excluded).Upon further examination, one record [44] was also removed by following the criteria from by Bauer and colleagues [32]. The study invokes intersectionality but draws a conventional category-by-category comparison where contrasts are made by comparing within poverty levels and within ethnic groups separately. The full screening process is outlined in Fig 1, and the full research results are included as supplementary information (S1 Appendix).

## Summary of studies included in review

The final review included one study from modelled projections [31] and three cross-sectional studies [41–43]. Common sociodemographic factors across all studies included ethnicity/race, deprivation (i.e., education, income), and gender/sex. All four studies conducted both single-categorical and intersectional analyses to tease out any underlying intersectionality effects. Their characteristics, methods, and key findings are summarised in Table 2 and Table 3, respectively.

## MLTC and life expectancy (Lam et al. (2024))

This projection-modelling study [31] used individual-level panel data from 5 waves of South African national surveys to model projections on life expectancy and multimorbid life expectancy at the intersection of ethnicity and education. In single-axes analyses, both higher overall life expectancy and multimorbid life expectancy were projected to be higher among White individuals and Asian/Indian individuals over mixed ethnicities and African individuals; those with higher levels of education over lower levels of education; and women over men. Intersectional analyses revealed a more nuanced pattern. Those with lower levels of education were projected to experience a higher proportion of multimorbid life expectancy across all ethnicities and sexes. In contrast, the intersectional modelling results did not show substantial differences in projected overall life expectancy by ethnic group.

## MLTC prevalence and 5-year all-cause mortality (Moreno-Juste et al. (2023), Moreno-Juste et al. (2024))

Two cross-sectional studies from Aragon, a region of north-eastern Spain, explored the role of intersectionality among gender, migration status, residence area, and income on MLTC prevalence [41] and all-cause mortality [42]. Moreno-Juste and colleagues (2023) examined MLTC prevalence among those with at least one LTC [41]. In single-categorical analyses, MLTC prevalence was higher for those with low income, being a woman, and living in urban areas, but lower for having been a migrant for less than 15 years. The authors attributed the latter to the healthy immigrant paradox. Intersectional modelling also found MLTC prevalence to be higher among women and those with lower income across all intersectional strata. The study concluded that, among those with at least one LTC in Aragon, Spain, lower income levels may impose a greater adverse impact for MLTC prevalence on women than men. Moreno-Juste and colleagues (2024) also examined 5-year all-cause mortality among those with MLTC [42] from the same EpiChron cohort of Aragon, Spain. The single-categorical analyses found mortality to be higher among those who were male, in a low-income group, and living in rural areas, but lower for having been a migrant for less than 15 years. Intersectional modelling found mortality inequalities at the intersections of income and gender, income and migration status, and residence area and gender. But here, men in the intersectional strata of lower income with in rural residence area, regardless of migration status, showed the worst mortality outcomes.

## MLTC-related emergency visits, hospitalisation, and mortality (Zandam, Akobirshoev, and Mitra (2024))

A cross-sectional study from the United States [43] examined disparities in MLTC-related emergency visits, hospitalisation, and mortality at the intersections of ethnicity and disability. Single-categorical analyses suggested that those with disability had higher rates of adverse MLTC-related outcomes than non-disabled individuals, and that Black and Latin individuals had higher rates of adverse outcomes than White individuals. Intersectional analyses results pointed to ethnicity

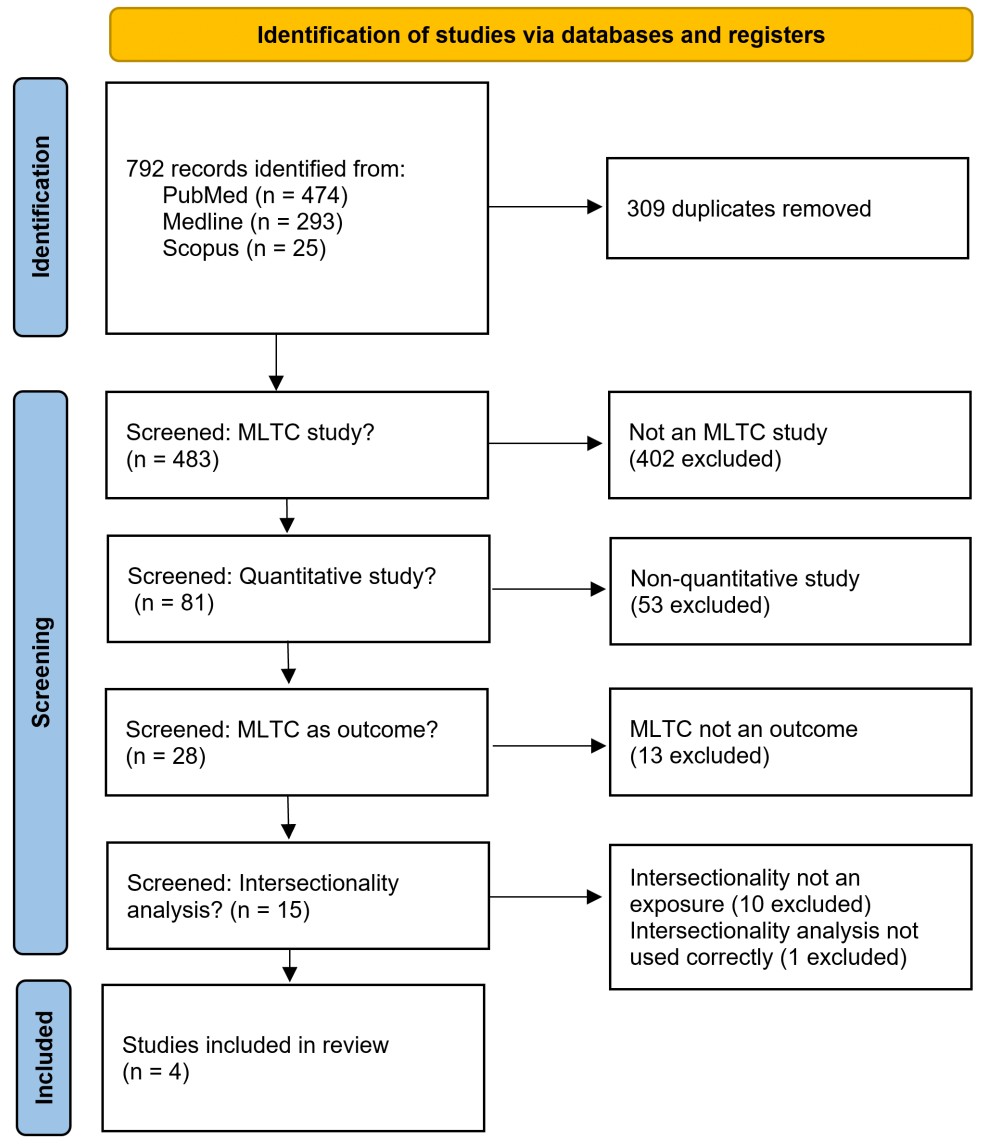

Source: Page MJ, et al. BMJ 2021;372:n71. doi: 10.1136/bmj.n71.

**Fig 1. The screening process flowchart for the search results.**

playing a role of compounding disadvantage on MLTC-related emergency visits, hospitalisation and mortality when intersected with disability, i.e., higher risks among Black and Latin individuals with disability than White individuals with disability for all three types of outcomes.

## Risk of bias among contributing studies

The final risk of bias assessment results are included in Table 3, and details reported in S2 Table. Note that the definition and selection of controls in this review was based on those with fewer LTCs, rather than a "healthy" control group (no

**Table 2. Summary of study characteristics.**

| Study | Moreno-Juste et al. (2023) | Moreno-Juste et al. (2024) | Zandam, Akobirshoev, and Mitra (2024) | Lam et al. (2024) |
|---|---|---|---|---|
| **Study design** | Cross-sectional | Cross-sectional | Cross-sectional | Longitudinal |
| **Data source** | EpiChron Cohort: electronic health records from primary and hospital health care, pharmacy billing records, and users' database | EpiChron Cohort: electronic health records from primary and hospital health care, pharmacy billing records, and users' database | 2020 Healthcare Cost and Utilisation Project – Nationwide Emergency Department Sample | 5 waves of panel survey data from the South African National Income Dynamics Study (NIDS) 2008–2017 |
| **Exposure** | Gender, migration status and length of stay in Aragon, residence area and annual gross income (proxy for socioeconomic class) | Gender, migration status and length of stay in Aragon, residence area and annual gross income (proxy for socioeconomic class) | Disability, ethnicity, age, type of health insurance, median area household income (proxy for socioeconomic class), hospital location, hospital teaching status, hospital region | Age, residence, ethnicity, education (proxy for socioeconomic class) |
| **Intersecting factors** | 36 possible permutations of all exposure variables | 24 possible permutations of all exposure variables | Ethnicity (White, Black, Latin, other/mixed) x disability | Ethnicity (African, Asian/Indian, mixed, White) x education |
| **Outcome** | MLTC prevalence (age-adjusted odds ratios across 36 intersectional strata) | All-cause mortality (age-adjusted odds ratios across 24 intersectional strata; cause of death was not available) | MLTC-related emergency department (ED) visits, hospitalisation following MLTC-related ED visits, and mortality during MLTC-related ED visits (risk ratios adjusted for exposure variables) | Life expectancy and multimorbid life expectancy (both at age 20 and up to age 85, adjusted for age, residence, and interaction groups) |
| **Reference group** | High-income, non-migrant, urban men | Middle-high income, non-migrant, urban women | White, non-disabled | White, well-educated |
| **Population** | Non-general population - All patients who presented with at least one LTC (n = 1,038,307) during January-December 2019 | Non-general population - All patients who presented with MLTC (n = 652,201) in 2015, followed up for 5 years (2016–2020) | Quasi-general population - Adults aged 18–64 years with a 3:1 control-case ratio of non-disabled adults (n = 296,394) and disabled adults (n = 99,538) | General population - Adults (n = 18,030) from 5 waves (51% in 5 waves, 20% in 4 waves, 13% in 3 waves, 16% in 2 waves) |
| **Notes on population** | Most of the study population were Spanish natives (87.4%), lived in urban areas (60.1%), and had a low annual gross income (71.4%). | Most of the study population were Spanish natives (89.8%), older than 45 years (77.9%), and had a low annual gross income (69.1%). | Excluded observations that had diagnoses of vision, hearing, and physical disability that did not overlap with observations with intellectual and developmental disabilities (IDD) | The baseline sample was collected using a two-stage clustered sampling design and consisted of over 28,000 individuals from more than 7,300 households |
| **MLTC definition** | 2 or more LTCs among 153 conditions present, identified, and consolidated from within the electronic health records from Aragon, Spain | 2 or more LTCs among 129 conditions present, identified, and consolidated from within the electronic health records from Aragon, Spain | 2 or more LTCs among 20 LTCs set by the Agency for Healthcare Research and Quality (AHRQ) definition | 2 or more LTCs among 14 chronic diseases which include HIV and tuberculosis to reflect the context of South Africa |
| **Location** | Spain (Aragon) | Spain (Aragon) | United States (nationwide) | South Africa |

history of disease) originally intended for single-disease, single-outcome studies. The risk of bias for all studies included in the final review was assessed to have a satisfactory level of methodological quality overall.

## Discussion

MLTC builds along the cumulative disadvantages experienced across the life course [13,31], starting with adverse childhood experiences and socioeconomic conditions [7,45]. However this review includes three cross-sectional studies [41–43], two of which [41,42] originated from the same cohort, and the fourth study was based on modelled projections

**Table 3. Summary of key findings.**

| Study | Moreno-Juste et al. (2023) | Moreno-Juste et al. (2024) | Zandam, Akobirshoev, and Mitra (2024) | Lam et al. (2024) |
|---|---|---|---|---|
| **Type of outcome** | MLTC Prevalence | 5-year all-cause mortality | MLTC-related emergency visits, hospitalisation, and mortality | Years expected to live at age 20 (up to age 85), including the years lived with MLTC |
| **Key single-categorical findings** | MLTC prevalence increases by:<br>- Having low income<br>- Being a woman<br>- Living in urban areas<br>But decreases by:<br>- Being a migrant, especially for ≤15 years | Overall mortality likelihood increases by:<br>- Being male<br>- Being in a low-income group<br>- Living in rural areas<br>But decreases by:<br>- Being a short-term migrant (≤15 years) | - Across all ethnic groups, those with disability had higher rates of emergency visits, hospitalisation, and mortality than non-disabled even after adjusting for sociodemographic and hospital characteristics.<br>- Non-disabled Black individuals and Latin individuals had higher rates of emergency visits, hospitalisation, and mortality than non-disabled White individuals. | Both higher overall life expectancy and multimorbid life expectancy are higher among<br>- White individuals and Asian/Indian individuals over mixed ethnicity and African individuals;<br>- Those with post-secondary education over lower levels of education; and<br>- Females over males. |
| **Key intersectional findings** | Even after a full intersectional analysis modelling and adjusting for age, the likelihood of MLTC ran along the gradients of income and sex (women). | Among those with MLTC, all-cause mortality risk was higher in men than women regardless of income, migration status, and residence area, and especially among men with low income and in those living in rural areas. | The role of disability and ethnic minority status combined were compounding, i.e., higher risks among Black individuals and Latin individuals with disability than White individuals with disability for all 3 outcomes of interest. | - For both sexes and all ethnicities, those with less education experience a higher proportion of multimorbid life expectancy;<br>- No substantial difference in overall life expectancy per ethnicity, except among the mixed ethnicity group. |
| **Key insights** | In terms of MLTC prevalence for those with at least one LTC, lower income levels may exhibit a greater impact on women than men. | Significant inequalities of overall mortality among those with MLTC were found at the intersections of gender and income, gender and residence area, and migrant status and income. | For MLTC-related emergency visits, hospitalisation, and deaths, disability is a compounding risk factor across all ethnic groups. | For both overall life expectancy and multimorbid life expectancy, no clear evidence of cumulative disadvantages was apparent at the intersection of ethnicity and education for both sexes. |
| **Analytical method** | Consecutive logistic regression models examining disparities in MLTC prevalence in the following order:<br>(1) sex<br>(2) model (1) + income<br>(3) model (2) + ethnicity and length of stay in Aragon<br>(4) model (3) + residence area<br>and<br>(5) model (4) + 36 intersectional strata formed from all possible permutations of four variables | Consecutive logistic regression models examining disparities in MLTC 5-year all-cause mortality in the following order:<br>(1) sex<br>(2) model (1) + income<br>(3) model (2) + ethnicity and length of stay in Aragon<br>(4) model (3) + residence area and<br>(5) model (4) + 24 intersectional strata formed from all possible permutations of four variables | Multilevel Poisson regression to examine disparities in MLTC-related emergency visits, hospitalisation, and mortality at the intersections of ethnicity and disability, adjusted for socio-demographic factors | (1) Incidence-based Markov model estimates how much of projected life expectancy will be spent healthy, multimorbid, or in transitions between states;<br>(2) multinomial logit regression to examine how transition probabilities differ at the intersections of ethnicity and education, adjusted for socio-demographic factors |
| **Intersectional analysis category** | - Regression modelling with interaction terms<br>- Area under the receiver operating curve (AUROC) | - Regression modelling with interaction terms<br>- Area under the receiver operating curve (AUROC) | Regression stratified by subgroups | Regression with interaction terms |
| **Risk of bias** | Low | Low | Low | Low |

[31]. It limits our ability to examine the effect of intersectional effects on MLTC outcomes over the life course from this review, for example the age of index condition onset [46–49]; temporal order of MLTC accumulation [50–53]; transition probabilities between MLTC states [52,54,55]; or speed of MLTC accumulation [49]. The limited evidence in this review, however, may still suggest that MLTC outcomes can vary considerably at the intersections of sociodemographic factors.

The role of income on MLTC outcomes can vary by what other sociodemographic factors it intersects with. A low level of income is often identified as a key MLTC risk factor [6,7,15,16]. However, the two studies from Aragon, Spain [41,42] found that the role of income on MLTC outcomes may not be equal for everyone. Here, lower income levels associate with higher MLTC prevalence in women than men, while mortality outcomes were consistently worse in men than women across all income levels, migration status, and residence area. It is also interesting to note that even the single-categorical analyses point to rural residence being associated with lower MLTC prevalence but higher mortality.

The deep racial divisions within the US context may be the driver behind the role of disability on MLTC outcomes varying when intersected with ethnicity. Past systematic reviews identified both disability [7] and being an ethnic minority [16] as risk factors for worse MLTC related outcomes. The US-based study included in this review suggests that, as Crenshaw's earlier writing [29] on the intersectional experience of discrimination for American Black women reads, the experience of disability-related MLTC inequality may depend very much on the racial context of each individual.

A low level of education is a known MLTC risk factor [6,15,16,56], but when intersected with ethnicity for both men and women in the South African setting, definitive cumulative disadvantages were not found in the projected life expectancy [31]. Here, the legacy of the historical disparities in South Africa may remain so deeply entrenched between ethnicity and socioeconomic status [57] to hinder the usual protective role of higher education against adverse health outcomes, i.e., MLTC, for non-White ethnicities.

The morbidity-mortality paradox, where "females live longer than males but spend a higher proportion of their total life expectancy in poorer health states" [58], was partially observed in three [31,41,42] studies where MLTC outcome disparities were examined by sex. However, it remains challenging to postulate whether these findings are generalisable without a deeper exploration of intersectional contexts of each study and further evidence besides the limited findings of this review.

### Limitations of the evidence included in the review

Besides the lack of longitudinal evidence, the small volume of evidence is a key limitation of this review. It may be because the quantitative application of intersectionality to health research is still in infancy [32,35,39], and intersectionality-focused longitudinal MLTC research would demand access to both high computational power and large, high-quality longitudinal datasets, limiting the number of studies available.

Another key limitation of the existing evidence is the lack of solution-linked variables. Based on the original definition [59], solution-linked variables are variables which can be modified or intervened upon to mitigate intersectional inequalities in MLTC trajectories. However, like many other intersectionality-focused quantitative studies [60], the studies included in this review do not include such variables. Continuing to analyse inequalities in increasingly finer, intersectional details without suggesting actionable solutions may strengthen ideas that health inequalities are inherently natural among groups [32].

### Conclusions

This systematic review highlights the importance of applying an intersectional framework to MLTC research to better understand disparities. The limited evidence may potentially suggest that, depending on the relational context, MLTC outcomes can vary considerably at the intersections of sociodemographic factors. However, it is too early to draw a definite conclusion from a very small body of non-longitudinal evidence.

Future research should prioritise studying MLTC trajectory with an intersectionality lens using longitudinal data (e.g., large-scale electronic health records or individual-level panel surveys that ensure a diverse population) to examine how

intersecting disadvantages accumulate onto outcome disparities over time. Moreover, MLTC-intersectionality longitudinal study models should identify and include solution-linked variables to inform more context-responsive MLTC interventions.

## Supporting information

**S1 Table.  PRISMA 2020 checklist.**
(DOCX)

**S2 Table.  Newcastle-Ottawa Scale risk of bias of reviewed studies.**
(DOCX)

**S1 Appendix.  Search results from 16 April 2025.**
(XLSX)

## Author contributions

**Conceptualization:** Mansuk Daniel Han, Thomas Yates, Francesco Zaccardi.

**Data curation:** Mansuk Daniel Han.

**Formal analysis:** Mansuk Daniel Han.

**Investigation:** Mansuk Daniel Han.

**Methodology:** Mansuk Daniel Han, Thomas Yates, Francesco Zaccardi.

**Project administration:** Mansuk Daniel Han.

**Supervision:** Thomas Yates, Kamlesh Khunti.

**Validation:** Francesco Zaccardi.

**Visualization:** Mansuk Daniel Han.

**Writing – original draft:** Mansuk Daniel Han.

**Writing – review & editing:** Mansuk Daniel Han, Thomas Yates, Kamlesh Khunti, Cameron Razieh, Francesco Zaccardi.

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
