## [Decision Letter · Decision Letter 0]

21 Jul 2025

Dear Dr. Han,

Thank you for submitting your manuscript to PLOS ONE. After careful consideration, we feel that it has merit but does not fully meet PLOS ONE’s publication criteria as it currently stands. Therefore, we invite you to submit a revised version of the manuscript that addresses the points raised during the review process.

We look forward to receiving your revised manuscript.

Kind regards,

Filipe Prazeres, MD, MSc, Ph.D.

Academic Editor

PLOS ONE

Journal Requirements:

[KK is supported by the National Institute for Health Research (NIHR) Applied Research Collaboration East Midlands (ARC EM), NIHR Global Research Centre for Multiple Long Term Conditions, NIHR Cross NIHR Collaboration for Multiple Long Term Conditions, NIHR Leicester Biomedical Research Centre (BRC) and the British Heart Foundation (BHF) Centre of Excellence.]

[The author(s) received no specific funding for this work.]

3. Please include captions for your Supporting Information files at the end of your manuscript, and update any in-text citations to match accordingly. Please see our Supporting Information guidelines for more information: http://journals.plos.org/plosone/s/supporting-information .

Reviewers' comments:

Reviewer's Responses to Questions

**Comments to the Author**

1. Is the manuscript technically sound, and do the data support the conclusions?

Reviewer #1: Partly

Reviewer #2: Yes

2. Has the statistical analysis been performed appropriately and rigorously?

Reviewer #1: No

Reviewer #2: Yes

3. Have the authors made all data underlying the findings in their manuscript fully available?

Reviewer #1: Yes

Reviewer #2: Yes

4. Is the manuscript presented in an intelligible fashion and written in standard English?

Reviewer #1: Yes

Reviewer #2: Yes

Reviewer #1: Dear Authors,

Thank you for the opportunity to review your manuscript. This systematic review addresses an important and underexplored topic in the field of public health — the role of sociodemographic intersections in the development and progression of multiple long-term conditions (MLTC). Your effort to apply an intersectional lens to the issue of multimorbidity is timely and commendable.

However, I believe that substantial revisions are required to improve the clarity, theoretical foundation, and methodological transparency of your work. Below are my detailed comments and suggestions for each section of the manuscript:

1. Abstract

The abstract appropriately outlines the study objectives, methods, and conclusions. However, it could benefit from a clearer statement of the main findings and their implications.

Please clarify that only four studies were included, and emphasize the limitations upfront to better align expectations for the reader.

2. Introduction

The concept of intersectionality is introduced, but the theoretical framework requires further development. I suggest incorporating key foundational references (e.g., Crenshaw, Bowleg) and clarifying how intersectionality differs from statistical interaction or additive models.

The objective of the review should be more explicitly stated at the end of the introduction. Consider adding a sentence such as: “The aim of this systematic review is to…”

3. Methods

The eligibility criteria are well defined; however, the rationale for excluding studies with specific populations (e.g., people with diabetes) could be better justified.

The description of how intersectional approaches were evaluated in the included studies lacks clarity. Please elaborate on the criteria used to determine whether studies “correctly” applied intersectionality analyses.

The use of the PICO framework is unconventional for this type of review. The term “control” as “unidimensional factors” is confusing and may not be necessary.

4. Results

The presentation of the four included studies is informative, and the tables are helpful. However, given the very limited number of included studies — with two from the same cohort — the generalizability of findings should be presented more cautiously.

In several parts of the text, findings are overstated given the small and heterogeneous evidence base. I recommend using more qualified language such as “may suggest” or “indicate a potential trend.”

5. Discussion

The discussion offers a useful synthesis but occasionally draws strong conclusions from limited evidence.

Please be more explicit about the implications of relying primarily on cross-sectional data and modeled projections.

The section on “solution-linked variables” is important but underdeveloped. Consider expanding this discussion to provide concrete examples of variables that could be used in future research or interventions.

The section on the morbidity-mortality paradox is interesting but could be shortened or more tightly linked to the intersectional focus of the study.

6. Limitations

I appreciate your candid acknowledgment of limitations. However, the limitation regarding publication language bias could be removed or softened unless you found evidence that it affected your results.

The limitation that three of the four studies are cross-sectional should be emphasized more strongly in relation to the study’s objectives.

7. Conclusion

The conclusion accurately summarizes the study but could be strengthened by clearly stating what future research should prioritize (e.g., longitudinal data, use of solution-linked variables, inclusion of diverse populations, theory-informed methods).

8. Writing Style and Structure

Overall, the manuscript is well written, but some sections are overly dense and could be streamlined to improve readability.

There are some redundancies between tables and narrative descriptions. Consider trimming repetitive text.

Final Remarks

This manuscript addresses a critical and timely issue in global health. With revisions to the conceptual framing, methodological explanations, and cautious interpretation of results, it has the potential to contribute meaningfully to the literature on MLTC and health inequities.

Thank you again for your contribution, and I hope my suggestions are helpful in strengthening your manuscript.

Best regards,

Reviewer #2: Peer Review – Systematic Review Article

General Assessment:

The manuscript submitted presents the original version of a systematic review. The work is articulated in clear and accessible language, while maintaining scientific rigor. The title is appropriately informative, and the abstract provides a satisfactory summary of the study's scope, methods, and main findings.

Methodological Evaluation:

The authors thoroughly reported the methodology, adhering to the PICO framework. All key stages of the systematic review process were clearly described, including:

• Detailed search strategies across scientific databases, registries, and relevant websites;

• Definition of outcomes and description of data collection procedures;

• Transparent explanation of the processes used to determine the eligibility of studies for each synthesis;

• Explicit clarification of the number of reviewers involved in the selection of records and full-text articles.

The characteristics of the included studies were systematically summarized, and the risk of bias across studies was assessed and reported. The manuscript presents the statistical syntheses in a comprehensive manner, including interpretation of results consistent with the evidence retrieved.

Critical Analysis and Interpretation:

The discussion section appropriately addresses the limitations of both the included evidence and the review methodology. The implications of the findings for clinical practice, health policy, and future research are well articulated.

The review also includes registration and protocol information, contributing to its transparency and reproducibility.

Conclusion:

Overall, the authors provided a complete and methodologically sound description of the processes employed to identify, select, appraise, and synthesize the studies. The manuscript meets the standards expected for systematic reviews.

Recommendation:

I congratulate the authors for the quality and clarity of the work. I have no further suggestions at this time.

**Do you want your identity to be public for this peer review?** For information about this choice, including consent withdrawal, please see our Privacy Policy

Reviewer #1: **Yes: ** Bruno Holanda Ferreira

Reviewer #2: No

---

## [Author Response · Author response to Decision Letter 1]

17 Sep 2025

1. Abstract: Please clarify that only four studies were included, and emphasize the limitations upfront to better align expectations for the reader.

- Revised abstract now includes: “Four studies met inclusion criteria but none were longitudinal, which limits our ability to examine the role of intersectional effects on MLTC outcomes over the life course from this review. A narrative synthesis was conducted due to their wide methodological heterogeneity. The limited evidence may potentially suggest that MLTC outcomes can vary considerably at the intersections of sociodemographic factors.”

2. Introduction: The concept of intersectionality is introduced, but the theoretical framework requires further development. I suggest incorporating key foundational references (e.g., Crenshaw, Bowleg) and clarifying how intersectionality differs from statistical interaction or additive models.

- Agreed; reworked and expanded

- Delved back into the literature on intersectionality theory [1, 2], quantitative application of intersectionality in public health [3-7], and critique on the social determinants of health [8, 9]

The objective of the review should be more explicitly stated at the end of the introduction. Consider adding a sentence such as: “The aim of this systematic review is to…”

- Revised: The aim of this systematic review is to examine how the interplay of sociodemographic factors plays a role on MLTC outcomes by their relational context. We identify and summarise quantitative studies that examine MLTC outcomes through an intersectionality lens; assess which social positions (e.g. gender, ethnicity, income) have been studied in combination; and discuss how their interplay in MLTC outcomes varies when considered within an intersectional context.

3. Methods: The eligibility criteria are well defined; however, the rationale for excluding studies with specific populations (e.g., people with diabetes) could be better justified.

- Removed the criterion upon further review.

- Review results remain the same: In retrospect what we actually had done was not restricted to the ‘general population’. For example, two of the four studies included in the final synthesis examine populations with at least one long-term condition [10] or already with multimorbidity [11].

- Theoretical justification: Current literature reveal that there is no rationale to restrict ourselves to a ‘general’ condition, because one of the key components of an MLTC trajectory includes the type of index condition [12-17].

The description of how intersectional approaches were evaluated in the included studies lacks clarity. Please elaborate on the criteria used to determine whether studies “correctly” applied intersectionality analyses.

- Revised paragraph: Particular attention was paid to whether the studies applied intersectionality analysis approaches correctly. Because the use of intersectionality approach in quantitative health research is in nascent stages [6, 18, 19], the claims of applying intersectionality in quantitative health research warrant case-by-case examination. Therefore, we examined each study by the criteria for improving quantitative intersectional research suggested by Bauer and colleagues [7].

- Revised the ‘study selection’ subsection: Upon further examination, one record [20] was also removed by following the criteria from by Bauer and colleagues [7]. The study invokes intersectionality but draws a conventional category-by-category comparison where contrasts are made by comparing within poverty levels and within ethnic groups separately.

The use of the PICO framework is unconventional for this type of review. The term “control” as “unidimensional factors” is confusing and may not be necessary.

- Agreed; removed Table 1 and associated text.

4. Results: The presentation of the four included studies is informative, and the tables are helpful. However, given the very limited number of included studies — with two from the same cohort — the generalizability of findings should be presented more cautiously. In several parts of the text, findings are overstated given the small and heterogeneous evidence base. I recommend using more qualified language such as “may suggest” or “indicate a potential trend.”

- Agreed, softened the language with more nuances where applicable.

5. Discussion: The discussion offers a useful synthesis but occasionally draws strong conclusions from limited evidence.

- Agreed, softened the language with more nuances where applicable.

Please be more explicit about the implications of relying primarily on cross-sectional data and modeled projections.

- Revised 1st paragraph: MLTC builds along the cumulative disadvantages experienced across the life course [21, 22], starting with adverse childhood experiences and socioeconomic conditions [23, 24]. However this review includes three cross-sectional studies [10, 11, 25], two of which [10, 11] originated from the same cohort, and the fourth study was based on modelled projections [22]. It limits our ability to examine the effect of intersectional effects on MLTC outcomes over the life course from this review, for example the age of index condition onset [26-29]; temporal order of MLTC accumulation [14-17]; transition probabilities between MLTC states [16, 30, 31]; or speed of MLTC accumulation [29]. The limited evidence in this review, however, may still suggest that MLTC outcomes can vary considerably at the intersections of sociodemographic factors.

The section on “solution-linked variables” is important but underdeveloped. Consider expanding this discussion to provide concrete examples of variables that could be used in future research or interventions.

- Revised: Another key limitation of the existing evidence is the lack of solution-linked variables. Based on the original definition [32], solution-linked variables are variables which can be modified or intervened upon to mitigate intersectional inequalities in MLTC trajectories. However, like many other intersectionality-focused quantitative studies [33], the studies included in this review do not include such variables. Continuing to analyse inequalities in increasingly finer, intersectional details without suggesting actionable solutions may strengthen ideas that health inequalities are inherently natural among groups [7].

The section on the morbidity-mortality paradox is interesting but could be shortened or more tightly linked to the intersectional focus of the study.

- Agreed, we cut out the superfluous bits in the middle.

- Revised: The morbidity-mortality paradox, where “females live longer than males but spend a higher proportion of their total life expectancy in poorer health states” [34], was partially observed in three [10, 11, 22] studies where MLTC outcome disparities were examined by sex. However, it remains challenging to postulate whether these findings are generalisable without a deeper exploration of intersectional contexts of each study and further evidence besides the limited findings of this review.

6. Limitations: I appreciate your candid acknowledgment of limitations. However, the limitation regarding publication language bias could be removed or softened unless you found evidence that it affected your results

- No evidence that it affected the results; therefore removed.

The limitation that three of the four studies are cross-sectional should be emphasized more strongly in relation to the study’s objectives.

- See the revised 1st paragraph of Discussion above.

7. Conclusion: The conclusion accurately summarizes the study but could be strengthened by clearly stating what future research should prioritize (e.g., longitudinal data, use of solution-linked variables, inclusion of diverse populations, theory-informed methods)

- Revised: This systematic review highlights the importance of applying an intersectional framework to MLTC research to better understand disparities. The limited evidence may potentially suggest that, depending on the relational context, MLTC outcomes can vary considerably at the intersections of sociodemographic factors. However, it is too early to draw a definite conclusion from a very small body of non-longitudinal evidence…Future research should prioritise studying MLTC trajectory with an intersectionality lens using longitudinal data (e.g. large-scale electronic health records or individual-level panel surveys that ensure a diverse population) to examine how intersecting disadvantages accumulate onto outcome disparities over time. Moreover, MLTC-intersectionality longitudinal study models should identity and include solution-linked variables to inform more context-responsive MLTC interventions.

8. Writing Style and Structure: Overall, the manuscript is well written, but some sections are overly dense and could be streamlined to improve readability. There are some redundancies between tables and narrative descriptions. Consider trimming repetitive text.

- Agreed; performed major streamlining across the whole manuscript.

Journal Requirements

- Formatted the title page accordingly, e.g. UK > United Kingdom and NIHR > National Institute for Health and Care Research, per ‘no abbreviation’ rule

- Headings are fixed to level 1 (pt 18, bold), level 2 (pt 16, bold), level 3 (pt 14, bold)

- In-line Vancouver-style citations are now in square brackets, e.g. “There are modifiable and non-modifiable MLTC risk factors [5, 9]. Individual lifestyle and behavioural attributes [9, 10] are considered modifiable factors and intervention targets, such as stress, sleep, physical activity, smoking, or diet [11].”

- Changed file names to: “Fig1.tif”, “S1_Table.docx”, “S2_Table.docx”, “S3_Appendix.xlsx”

2. Please remove any funding-related text from the manuscript and let us know how you would like to update your Funding Statement.

- Removed all “Funding” related texts from the manuscript as advised

- Indicated on the rebuttal letter that we request PLoS One change the Funding Statement section of the online submission form on our behalf, from current “The author(s) received no specific funding for this work”, to…

o The Leicester Real World Evidence Unit is supported by the National Institute for Health and Care Research (NIHR) Applied Research Collaboration East Midlands (ARC EM) and the NIHR Leicester Biomedical Research Centre (BRC).

o KK is supported by the National Institute for Health Research (NIHR) Applied Research Collaboration East Midlands (ARC EM), NIHR Global Research Centre for Multiple Long Term Conditions, NIHR Cross NIHR Collaboration for Multiple Long Term Conditions, NIHR Leicester Biomedical Research Centre (BRC) and the British Heart Foundation (BHF) Centre of Excellence.

3. Please include captions for your Supporting Information files at the end of your manuscript, and update any in-text citations to match accordingly.

- Changed the file names to, S1 Table, S2 Table, and S3 Appendix; and changed in-text citations to match accordingly:

o “S1 Table for the PRISMA 2020 checklist”

o “The final risk of bias assessment…details reported in S2 Table.”

o “The full research results are included as supplementary information (S3 Appendix).”

---

## [Decision Letter · Decision Letter 1]

3 Nov 2025

Sociodemographic intersections and risk of multiple long-term conditions: a systematic review

PONE-D-25-21560R1

Dear Dr. Han,

We’re pleased to inform you that your manuscript has been judged scientifically suitable for publication and will be formally accepted for publication once it meets all outstanding technical requirements.

Kind regards,

Filipe Prazeres, MD, MSc, Ph.D.

Academic Editor

PLOS ONE

Additional Editor Comments (optional):

Reviewers' comments:

Reviewer's Responses to Questions

**Comments to the Author**

Reviewer #1: All comments have been addressed

2. Is the manuscript technically sound, and do the data support the conclusions?

Reviewer #1: Yes

3. Has the statistical analysis been performed appropriately and rigorously?

Reviewer #1: N/A

4. Have the authors made all data underlying the findings in their manuscript fully available?

Reviewer #1: Yes

5. Is the manuscript presented in an intelligible fashion and written in standard English?

Reviewer #1: Yes

Reviewer #1: Thank you for the opportunity to review the revised version of the manuscript entitled “Sociodemographic intersections and risk of multiple long-term conditions: a systematic review.” I appreciate the authors’ careful and thoughtful revisions, as well as the relevance and timeliness of the topic. The manuscript addresses an important gap in understanding how sociodemographic intersections influence multiple long-term conditions (MLTC), and the revised version shows substantial improvement in clarity, theoretical grounding, and alignment with intersectionality frameworks.

**Do you want your identity to be public for this peer review?** For information about this choice, including consent withdrawal, please see our Privacy Policy

Reviewer #1: **Yes: ** Bruno Holanda Ferreira

---

## [Editor Report · Acceptance letter]

PONE-D-25-21560R1

PLOS ONE

Dear Dr. Han,

I'm pleased to inform you that your manuscript has been deemed suitable for publication in PLOS ONE. Congratulations! Your manuscript is now being handed over to our production team.

Kind regards,

on behalf of

Prof. Filipe Prazeres

Academic Editor

PLOS ONE